# A Morphological and Morphometric Dental Analysis as a Forensic Tool to Identify the Iberian Wolf (*Canis Lupus Signatus*)

**DOI:** 10.3390/ani10060975

**Published:** 2020-06-03

**Authors:** Víctor Toledo González, Fernando Ortega Ojeda, Gabriel M. Fonseca, Carmen García-Ruiz, Pablo Navarro Cáceres, Pilar Pérez-Lloret, María del Pilar Marín García

**Affiliations:** 1Department of Analytical Chemistry, Physical Chemistry and Chemical Engineering, University of Alcalá, 28871 Alcalá de Henares (Madrid), Spain; fernando.ortega@uah.es (F.O.O.); carmen.gruiz@uah.es (C.G.-R.); 2University Institute of Research in Police Sciences (IUICP), University of Alcalá, 28801 Alcalá de Henares (Madrid), Spain; 3Department of Anatomy and Embryology, Faculty of Veterinary, Universidad Complutense of Madrid (UCM), 28040 Madrid, Spain; pilper01@ucm.es (P.P.-L.); pilmarin@vet.ucm.es (M.d.P.M.G.); 4Centro de Investigación en Odontología Legal y Forense (CIO), Facultad de Odontología, Universidad de La Frontera, Temuco 4780000, Chile; gabriel.fonseca@ufrontera.cl; 5Centro de Investigación en Ciencias Odontológicas, Facultad de Odontología, Universidad de La Frontera, Temuco 4780000, Chile; pablo.navarro@ufrontera.cl; 6Universidad Autónoma de Chile, Temuco 4780000, Chile

**Keywords:** forensics, analysis, Iberian wolf, bite marks, veterinary, dentistry

## Abstract

**Simple Summary:**

Attacks by Iberian wolves on farm animals routinely cause conflicts with humans and threaten their economic interests related to livestock. However, wolf predation can sometimes be confused with that caused by other carnivores like dogs. Some studies have tried to identify or differentiate canids as the predators responsible for such attacks by analysing their tooth/bite marks on bone remains. Nevertheless, most of those studies have only considered a few dental measurements, and they were carried out in a palaeoecological and zooarchaeological context. As there is still limited information on Iberian wolf‘s dental anatomy that can be used in forensic cases, this study aimed to describe the morphology of the Iberian wolf‘s teeth and to provide new morphometric characteristics, as complete as possible, to collaborate in the correct interpretation of a wolf‘s bite marks at crime scenes. Based on the morphometric dental analysis, it was possible to differentiate female and male wolves. Moreover, the dental morphometric characteristics described can be used, at least as a reference, to identify the Iberian wolf‘s tooth/bite marks or to rule out other potential aggressors.

**Abstract:**

Depredation by the Iberian wolf (*Canis lupus signatus*) is currently thought to be a problem in some areas of Spain. However, there are few technically validated forensic tools available to determine the veracity of claims with a high degree of scientific confidence, which is important given that such attacks may lead to compensation. The analysis of bite marks on attacked animals could provide scientific evidence to help identify the offender. Thus, the aim of this study was to assess the morphological and morphometric characteristics of Iberian wolf dentition. This data collection would serve as a base-point for a more accurate identification of the wolves thorough their bite marks. For the first time, 36 dental variables have been studied in wolves’ skulls, employing univariate and multivariate analyses. The general morphological dental characteristics of wolves are very similar in terms of their dental formula and tooth structure to other canids, like domestic dogs. Sex differentiation was evident, principally in terms of the maxillary distance between the palatal surfaces of the canine teeth (UbC) and the width of the left mandibular canine teeth (LlCWd). New morphometric reference information was obtained that can aid the forensic identification of bite marks caused by the Iberian wolf with greater confidence.

## 1. Introduction

Animals are capable of inducing severe injuries through bites, which can even result in death [1]. Indeed, some families of the order Carnivora include species that are capable of attacking and killing human beings. Wild animals only rarely kill in urban areas [2], with predation on livestock representing the main source of conflict between large carnivores and humans [3]. In such cases, it is important to be able to determine if the animal bites were the true cause of death, and when this is the case, to identify the perpetrator [1]. Among mammals, carnivores are the most common scavengers of human remains [4] and their post mortem outdoor scavenging activity has traditionally been the best documented from a forensic perspective [4,5,6]. The analysis of bite marks can be used by forensic experts to aid in the identification of the taphonomic agent, and to interpret the behaviour, scavenging patterns, and predatory conduct of different animal species [7]. 

Livestock attacks attributed to wolves are becoming more and more frequent in the Iberian Peninsula [8], producing important societal problems [9]. Wolf predation can be confused with that of other carnivores, mainly free-ranging domestic dogs, a phenomenon that is more widespread in southern European countries [10]. Confusing field conditions or insufficient technical skills often make it difficult to correctly identify the perpetrators of such attacks [11], highlighting the need to develop tools to identify predators [12]. Different scavenger species within the same family can have different tooth dimensions, bite forces, jaw muscle strengths, and scavenging behaviours and patterns, which may be affected by various other factors. These features affect the type of bite marks produced on bone surfaces [13,14], making it necessary to define the morphological and morphometric features of the dentition of animals like wolves in order to be able to resolve conflicts with farmers. Indeed, up until 2015 there were no specific data available on tooth morphometry in the wolf [15]. In addition to collecting information that may help identify an animal assailant, these data may aid in cases where animal scavenging has occurred. Considering the increase in the alleged attacks on livestock attributed to the Iberian wolf and the lack of a detailed dental information available for this species, this study sets out to characterize the complete dentition of the Iberian wolf. Accordingly, the morphological features of different dental series were characterised, performing a morphometric analysis of 21 maxillary and 15 mandibular bones from wolves of either sex.

## 2. Materials and Methods 

The dentition in the skull of 45 Iberian wolves (*Canis lupus signatus*) of both sexes (26 females and 19 males) was examined and measured. The skulls analysed in this study were obtained from the Mammalian Collection of the Spanish Natural Science Museum (Madrid, Spain). Only wolves skulls identified anatomically with mature dentition (permanent dentition), and with all teeth at eruption stage 3, were included in the study. This means that all incisors, premolars, and molar teeth had fully erupted into the occlusal plane, and that the visible cementoenamel junction was above the alveolus, in accordance with the code used by Geiger et al. [16]. No congenital skull anomalies or external dental damage were evident in the skulls sampled. All the measurements were made with a traceable digital calliper (Control Company, ISO 17025, Blacksburg, VA, USA) and registered in millimetres (mm).

### 2.1. Morphometric Study

Although the variables considered in this study were those based on Lemmons and Beebe‘s dental nomenclature, the arbitrary acronyms are used to simplify the visualization and data analysis (Table 1) [17]. 

The data presented here were obtained on three different days by two independent observers. For the LCW, LCc, LbC, L1PMc, and L1Mc, 15 non-articulated mandibles were positioned and attached to the cranium in order to make accurate measurements. The damaged or worn mandibular incisor teeth were not considered in this study for the morphometric analysis. Likewise, the maxillary series of molar teeth, and the second and third mandibular molar teeth were also not considered for the morphometric analysis. The UbC, LbC (not shown), and Ubi, and the canine width and length measurements (UrCWd, UrCLe, UlCWd, UlCLe (Figure 1A–D) and, LrCWd, LrCLe, LlCWd, LlCLe, Ur4PMtub, Ul4PMtub, Lr1Mtub, and Ll1Mtub), were taken from the furcation (the point at which the roots diverge) at the level of the cementoenamel junction (the boundary between dental enamel and cement: Figure 1D–F).

The UPi, UPC, and UP1PM measurements were taken from the mandibular symphysis (between both the central incisor teeth: Figure 1A, black circle), and the distal surface of the third incisors, canines and first premolars (PM1s) respectively, to the height of the alveolar bone (Figure 1C). Finally, only teeth with pointed cusps on the largest dental tubercles were considered for the UPM4c and L1Mc measurements.

### 2.2. Morphological Description

The external (crown) and internal (root) morphology of the wolf‘s teeth was described based on the anatomic nomenclature established in Lemmons and Beebe [17] and in Schaller [18]. Dental root X-rays were obtained at a veterinary clinic (Trasportix LW AL, portatile domiciliare radiografico digitale, TXLW-4kW model, Milan, Italy) using the following parameters: KVp 54–58; mA 50 and exposure 0.08 s. To illustrate the Iberian wolf bite mark, an impression of the maxillary and mandibular dental arch was registered on dental wax and photographed with a Nikon D60 Digital camera (Nikon, Tokyo, Japan), mounted on a Hama tripod (Hama Technics, S.L., Barcelona, Spain). The patterns were registered by pressing the maxillary bone into the dental wax up to the maxillary PM1 mark and the images were acquired at a natural 90° angle (perpendicular).

### 2.3. Statistical Analysis

A Kolgomorov–Smirnov test was initially performed to assess normality, followed by a parametric Student t-test test for independent samples and a non-parametric U Mann-Whitney test for independent samples. Statistical analyses were performed with the IBM SPSS software v23.0 (IBM-Corp, Armonk, NY, USA) using ANOVA with a factorial design and multiple comparisons. A 95% confidence interval was considered (*p* < 0.05) and the results are presented as intervals in millimetres (mm), with their mean and standard deviation (SD).

For the multivariate analysis, a principal components analysis (PCA) was used to explore the extent to which variables studied were related, and a partial least squares discriminant analysis (PLS-DA) was used to classify the data by sex (males and females) and to identify the critical variables that allow such differentiation. The multivariate analysis was performed with Soft Independent Modelling of Class Analogies (SIMCA) (Sartorius Stedim Biotech, Göttingen, Germany) [19] and seven cross-validation groups were used to consider similar observations in the same group, validating the calculated latent variables. The data were centred and scaled (Unit Variance, UV), and the software was set to calculate the boundaries with 95 % probability [19]. Only variables that presented statistically significant differences between the sexes (*p* < 0.05) were considered for PCA and PLS-DA analysis.

## 3. Results

### 3.1. Dental Morphology

The morphological patterns of the wolves’ teeth were described based on the observations presented in Figure 2, Figure 3 and Figure 4, in addition to the relevant bibliographic information [17,18]. The dentition of canids involves two different dental arches, each of which has its own characteristics and number of teeth defined as incisors, canines, premolars, and molars: the maxillary arch (Figure 1B,C) and the mandibular arch. Molar teeth were only detected in permanent dentition (dental maturity). Each tooth described had a visible part that projected into the mouth, called a crown, and a deeper part that extended into the mandibular, incisive and maxillary bones, the root. Enamel and cement usually cover the crown and root, respectively. Between the crown and the root, there is a mild constriction that constitute the neck or cementoenamel junction (the transition between the enamel of the crown and the cement of the root: Figure 1D–F). In general, the incisor, canine, premolar, and molar teeth have four exposed surfaces, and with a cusp or ridge constituting a fifth surface. The cusp is the point or the tip of the crown of a tooth. The surfaces of the teeth that face into the vestibule (the space between the lips and incisors/canine teeth, or between the cheek and the premolar/molar teeth) are the vestibular surfaces. Moreover, the space between the lips and incisors, and between the lips and the canine teeth is commonly described as the labial surface, while the space between the cheek and the premolar/molar teeth is known as the buccal surface. The dental surfaces that are orientated towards the tongue are described as the lingual surface, although for the maxillary teeth, this surface is often referred to as the palatal surface.

Different permutations of the anatomical nomenclature can be used to indicate the precise position of each structure. The occlusal surface is that of the premolar and molar teeth that faces their antagonist, and that contacts the teeth in the opposite jaw on closure, while the ridges or cusps of premolars that do not make contact are known as occlusal ridges. For the incisors, the ridge along the “occlusal surface” is referred to as the incisal ridge, a chisel-like shape with a sharp edge, and the cusp of the canine teeth is generally called the cusp surface. However, premolars and molars may have multiple cusps and tubercles. The surface that faces the adjacent teeth within the same dental arch is the contact surface, a “mesial” surface if it is directed forward towards the median line and a “distal” surface if it faces away from the median line of the face. In the case of the canine and some premolar teeth, the term “surface” is replaced by the term “edges” when these are sharp. The space between two teeth is referred to as the diastema.

#### 3.1.1. The Teeth of the Iberian Wolf

The full dentition of the adult Iberian wolf is comprised of 42 teeth of four different types: 12 incisor (I), 4 canine (C), 16 premolar (PM), and 10 molar (M) teeth (Figure 2A). Thus, the permanent dental formula of the Iberian wolf: 2(I3/3, C1/1, PM 4/4 and M2/3), where the first numbers above are for the maxillary teeth and the second are for the teeth of the mandibular bone. Given that the dentition is the same on both sides, the formula lists only one side enclosed in parentheses and it is to be multiplied by 2 to give the total number of teeth. 

##### Incisors

An analysis of the wolf’s incisors identified six maxillary and six mandibular incisors (three right and three left incisors) numbered as the first (I1), second (I2) and third (I3) incisor from the midline (Figure 1A–D, Figure 2A,B, and Figure 3B). Each incisor has one root (Figure 3B), with the three maxillary teeth implanted into the incisive bone (Figure 3A). Both the crowns and roots of the maxillary and mandibular incisors increase in size (length) from I1 to I3, as is also seen for the width of the crowns (the distance between the distal and mesial surface: Figure 3B). The maxillary I1 and I2 have three protrusions along the incisal edge called mamelons, with two lateral mamelons and one central mamelon (Figure 1A), while I3 is conical with no mamelons and a sharp cusp (Figure 2B). The central mamelon is the longest one. The labial vestibule surface of the maxillary incisors is longitudinally convex (Figure 2B) and lightly concave along most of their lingual surface. From a rostral view, the incisal ridge of I1 and I2 of the maxillary bone is slightly rounded while I3 presents an incisal ridge that is more pointed. The crown of the incisor teeth has a light downward angle of inflexion up to a few millimetres ventral to the cementoenamel junction. Moreover, the base of the crown of I3 is broad compared to I1 and I2. The mandibular incisors are smaller in size (length and width) than their maxillary counterparts. They have two mamelons with a similar curvature to that of the maxillary incisors, and they are angled obliquely forward and upwards. The incisal ridge is present in I1, I2, and I3 forming a sharp edge, and I1 and I2 are laterally compressed. The roots of the maxillary and mandibular I1 and I2 are straight, twice as long as the crown is high (Figure 3B, the mandibular incisors are not shown). The root and crown of I3 are arched, displaying a similar crown and root length (Figure 3B). The distal surface of I3 is sharp and it is possible to describe a soft sideways curvature of the root. The maxillary and mandibular I3s are laterally compressed.

##### Canines

An analysis of the wolf’s canines identified four canine teeth, each positioned behind I3, two maxillary and two mandibular canine teeth in each dental arch (one-right and one-left canine on the maxillary and mandibular bone: Figure 1A–D). These are the largest teeth in the Iberian wolf‘s mouth, all with a light convex mesial edge and a concave distal edge from the cementoenamel junction. The maxillary canines are wider and larger than the mandibular ones (Figure 2A,B). From the cementoenamel junction, the crowns in the maxillary canines project in a forward, downward, and slightly lateral direction (Figure 3A,C). By contrast, the mandibular canines project in a forward direction and upwards, with a clearer and larger lateral deviation than their maxillary counterparts (Figure 1D and Figure 2B). All canines present a single tip cusp, are conical with an elliptical to triangular cross-section, and the joint to their unique root is arched. The crown on the maxillary canine teeth is less curved than that on the mandibular canine teeth. Maxillary canines are larger than the mandibular ones. Like the incisors, the canine teeth have a longer root than their crown height and they are also laterally compressed.

##### Premolars

In this analysis, 16 premolar teeth were identified, eight in the maxillary dental arch (Figure 3A) and eight in mandibular dental arch (four-right and four-left premolars in the maxillary and mandibular bones: (Figure 1B–F and Figure 2A). The maxillary premolar teeth are positioned more laterally than the mandibular ones. The PM1 is the smallest of the series (Figure 2A) and the crown of the maxillary PM1 is slightly lower than its counterpart in the mandibular bone. The mandibular PM1 is the most rostral premolar tooth in the series and it has only one tubercle, with one tip cusp and one root (Figure 3C), a straight yet longer root relative to the crown: the root is twice the crown height. Both, mandibular and maxillary PM2s and PM3s are similar. These premolar teeth have two tubercles (mesial and distal) and two diverging roots (mesiobuccal and mesiopalatal: Figure 3A,C). The mesial edge of the maxillary crowns faces down and backwards, while in the mandibular ones the direction of the edge is upwards and backwards. By contrast, the caudal edge of PM2 and PM3 is directed downward and onward, and up and onwards, respectively. Both the mesial and caudal edge meet to form the tip cusp. The maxillary PM4 is a carnassial tooth with two tubercles. A transverse groove on the contact surface separates the bigger mesial tubercle from the smaller distal tubercle (Figure 1F and Figure 4A,B). The direction of the mesial edge is similar to that in PM2 and PM3. PM4 has three roots, two diverging roots belonging to the mesial tubercle (mesiobuccal and mesiopalatal) and one distal root belonging to the distal tubercle (Figure 3A).

##### Molar Teeth

Regarding the molar teeth, there were two right and two left molars in the maxillary bone (Figure 2A and Figure 3A) and three right and three left molars in the mandibular bone (Figure 3C), a total of five left and five right molar teeth. In both the maxillary and mandibular bones, the M1 is the largest of the series and it is a carnassial tooth (Figure 1E, Figure 3A,C). The maxillary M1 and M2 have three tubercles and three roots, and these three tubercles are distributed as two vestibular tubercles (mesiobuccal and distobuccal) and one smaller lingual tubercle (Figure 4B). The mesiobuccal tubercle is bigger than the distobuccal tubercle, and a deep groove separates these two vestibular tubercles. By contrast, an occlusion surface separates the distolingual tubercle from others. In terms of the roots, those from the vestibular tubercles diverge while the root of the distolingual tubercle travels to the palatine bone (hard palate). The mandibular M1 has three tubercles (mesial, intermediate, and distal: Figure 4A), of which the intermediate is the biggest. A transverse groove separates the first two and a small occlusion surface separates the smaller distal tubercle from the intermediate tubercle. Like the mandibular M2, this molar has two diverging roots (mesial and distal) that pertain to their respective tubercles. The mandibular M2 has two tubercles (mesial and distal) separated by a small occlusion surface. The mandibular M3 is the smallest of the series, and it has one tubercle and one root. Diastemas in the incisor, premolar, and molar teeth are usually of varying length or they may be absent.

#### 3.1.2. Normal Occlusion (the Closing of the Jaw)

In normal occlusion, the edges of the maxillary incisors cover those of the mandibular incisor apices, which means that the mandibular incisors come into contact with the lingual surface of the maxillary incisors, producing a scissor effect (Figure 2A,B). The maxillary I3s tooth interdigitates between the mandibular I3s and mandibular canines. The sharp mesial edges of the maxillary canines come into direct contact with the sharp distal edges of the mandibular canines (Figure 2A,B). The mandibular canines adopt a position caudal to the maxillary I3s and they come into direct contact with their sharp distal surface. The maxillary canine teeth adopt a position more lateral to the mandibular ones during occlusion (Figure 2A,B). The first two maxillary and last three mandibular premolars are interspersed. The more lateral position of the maxillary premolar teeth relative to the mandibular ones is more evident in occlusion. Finally, the molar teeth make contact along their occlusal surface. The occlusal surface of the maxillary M1 tooth comes into direct contact with the distal cusp that pertains to the distal tubercle of the mandibular M1 tooth. The distal cusp of the mandibular M2 tooth comes into direct contact with the occlusal surface of the maxillary M2, while the mandibular M1 tooth comes into direct contact with the lingual surface of the maxillary PM4 and M1. Finally, the mandibular M2 tooth contacts the lingual surface of the maxillaryM1 and M2.

When a wolves’ bite marks on dental wax were evaluated, the incisor and canine teeth form a curved arch on maxillary and mandibular jaw. The maxillary incisor and canine teeth are registered in the superficial (Figure 5A) and deep bite (Figure 5B). Figure 5C shows the maxillary incisor and canine teeth registered in the deep bite. In a deeper bite, the diameter of the PM4 and the canine bite marks appear to be slightly larger (Figure 5A vs. Figure 5B). Thus, the distance between the palatal (UbC) or lingual (LbC) surface of both canine teeth decreases with depth. A degree of tooth crowding is visible among the Iberian wolf‘s incisor teeth on both bones (maxillary and mandible bones), both in superficial and deep bite marks. Mandibular incisors bite marks are smaller than maxillary ones in deep marks.

The bite marks of the Iberian wolf’s mandibular premolars become progressively bigger towards the distal and maxillary PM4 (carnassial tooth), the latter the biggest of the series while the PM1 is the smallest. The maxillary bite mark is a little larger than the mandibular one, which explains the deeper bite. Premolar teeth have a straighter disposition in both the maxillary (not visualized) and mandibular arch (Figure 4A).

### 3.2. Morphometric Data

Although significant differences were detected among the morphometric variables, sex differentiation was only possible for UCW, LCW, UCc, LCc, UbC, LbC, UiW, Uic, Ubi, U1PMc, U4PMc, L1Mc, UP1PM, Ur1PMCc, Ul1PMCc, UrCWd, UrCLe, UlCWd, UlCLe, LrCLe, LlCLe, Ur4PMtub, Ul4PMtub, Lr1Mtub, and Ll1Mtub (*p* < 0.05). Of all the variables for which there were significant differences between the sexes (*p* < 0.05), the averages for males was larger than the mean measurements registered in females (Figure 6A–D). In addition, males reached larger minimal and maximum absolute values than females, although the variables in the females were more variable (Figure 6). However, the maximum absolute LrCle and LlCle values were larger in females and thus, they were considered as outliers (Figure 6C: variables that did not present a significant difference are not shown: *p* > 0.05).

We consider some of the observations as outliers (outside the 95 % confidence level), especially in females (Figure 6), which may explain why the maximum intervals for the LrCle and LlCle were larger in females than for males (Figure 6C). While the exclusion of these outliers from the statistical analysis affected some intervals, there were no significant changes in the average values of males and females, i.e., the relationship between sexes remained stable (Appendix A and Appendix A). In addition, when comparing the two sexes there were variables with no area of overlap and others with a differing degree of overlap (Figure 6). Accordingly, certain limits can be set for sex differentiation (Table 2).

In percentage terms, the mean UCW value was approximately 7.65% lower in females than in males, while the mean UCc value was 8.31% higher in males than females. The same variables registered on the mandible (LCW and LCc) were approximately 11% and 13% lower in females and males than the maxillary values. The mean values for UbC, LbC, Ur1PMCc, Ul1PMCc, UrCLe, UlCLe, LrCLe and LlCLe were 9–13% lower in females, and the mean values for the remaining variables that were significantly different (*p* < 0.05) were also lower in females (4–8%). There were no significant differences (*p* > 0.05) between the same variables measured on either side of the maxilla and mandible. However, they were different in absolute terms.

A PCA exploration allowed two relatively well separated groups with some overlap between them to be visualized (data not shown), and while the PLS-DA also showed a certain degree of overlap between the two groups, it allowed two well-defined groups to be discriminated. The relationship and degree of dispersion among the scores (specimens) within and between both groups was classified by sex. 

The exception was one specimen, a female skull, which seemed to share characteristics of both sexes. This unusual specimen was excluded from the multivariate analysis as it was considered an outlier (outside the 95 % confidence level) (Appendix A). This specimen exaggerates the high variability in the female group, yet its exclusion did not cause any variation in the distribution of the variables within and between the two sexes. Consequently, the three latent variables in Figure 7, t(1) t(2), and t(3), explained 51.1%, 9.13%, and 4.4 % of the model’s variability, respectively. Therefore, the model explained about 64.6% of the variance between the sexes. 

The contribution plot visualizes the variables and their contribution (weight) to the differentiation between sexes (Figure 8). The analysis showed that the most important variable for this differentiation was UrCWd (orange column), which presented the strongest variation between sexes. Sequentially, the next most important variables were UrCWd, UbC, UiW, LbC, U4PMc, UP1PM, and Ubi.

## 4. Discussion

In this study, we have performed a detailed analysis of the dentition of the Iberian Wolf, defining the different types of teeth, their morphology and morphometric features, as well as any sexual divergence in these parameters. An earlier study of 53 skulls from nine different Gray wolf sub-species (17 females, 21 males, and 14 indeterminate sex: Murmann et al. 2006 [2] took into account three dental measurements on the maxilla and two on the mandible, studying the MCW, maxillary and mandible tip (corresponding to the UCW, UCc, and LCc in our study, respectively). In this earlier study [2], the term mesial bone height (MBH) was employed for the measurement taken next to the most mesial portion of the canine between the canine teeth on the maxillary and mandibular bones. Here, we referred to this variable as the maxillary palatal intercanine surface neck height (UbC) and the mandibular lingual canine surface neck height (LbC), since the measurements were taken between palatal/lingual surfaces of the canine teeth, respectively, and not between their mesial surfaces. Nevertheless, that earlier study identified larger intervals for the UCW, UCc, LCc, UbC, and LbC parameters (adapted to our nomenclature). In the Iberian wolf, the limits of the LCc interval were lower than those recorded for Gray wolves and indeed, the minimum and maximum values of UCW, UCc, and LCc for female Iberian wolves were lower than those registered in Gray wolves. Finally, our UbC and LbC values were lower in both sexes. When these dental measurements were compared with those in different domestic dog breeds (*Canis familiaris*), they were generally smaller in the wolves [2]. However, a reliable comparison with our outcomes is not possible as the outcomes from the dog breeds were not classified by sex, breed or size, nor by sub-species, and no statistical analysis was performed. In light of the above, for reference purposes, we can only indicate that our results for UCW and UCc in male Iberian wolves largely overlapped with those of the nine sub-species of Gray wolves.

In addition to classifying our specimens, we used millimetres (mm) and not centimetres (cm) as the unit of measurement allowing us to obtain more precise results. Although the dog (*Canis lupus familiaris*) is a direct descendant of the Gray wolf [20,21], it displays greater morphological diversity than any other species, which makes it extremely difficult to compare between the two animals, especially when the data are not classified. This must be taken into consideration when performing more complete comparative studies to correctly differentiate between Iberian wolves and domestic dogs through dental analyses. Although the outcomes from this earlier study were broad and not sufficiently specific to the wolves’ dental morphometry, it did focus on canids and it established benchmarks for later efforts to identify offending animals thorough bite marks. In recent years, indirect dental morphometric studies have compared bite marks caused by wolves (American and Iberian wolves) and other species registered on various types of bone, and using different approaches [22,23,24]. It should be noted that the average size of the Iberian wolf lies somewhere between that of the American and European wolves [25].

The general dimensions (breadth and length), distribution and proportions of the classic tooth marks (pits, punctures, scores and furrows) have been described for wolves [22,24,26,27,28,29,30,31]. Similar studies have been performed on dogs and a few others have compared wolf’s and dog’s bite marks [2,32,33,34], yet none have assessed the distances between their teeth considering a large number of dental variables. Characteristic features of the animals and samples that could have influenced the measurements have been considered (e.g., overlapping), such as environmental factors. Canid dental morphology allows this species to be differentiated from other carnivores and characteristic post-mortem lesions caused by domestic dogs can now be recognized, reflecting the dog’s dental anatomy [10]. Most of these past studies on bite marks, focused principally on archaeological and zooarchaeological samples, and contexts, with ambiguous outcomes. This represented a huge effort undertaken by those authors whom worked with non-standardized samples, exposed to unknown external conditions, which may have caused the results variability they obtained. Indeed, they focused on bite mark characteristics caused by only a few teeth. To complement and extend these studies, here, we collected as many dental variables as possible in order to make it possible to identify the bite mark patterns caused by Iberian wolves in a forensic context. This inspired us to define dental reference values for the Iberian wolf, a possible aggressor or scavenger, which should help to identify its activity through the analysis of its bite marks. The skulls of domestic dogs are more varied than that of its ancestor, the Gray wolf, approximating to the variation found in the wild [35]. Some arch sizes can produce intercanine distances (ICDs), referred to as the U/LCc here, that vary as much as 70 mm in large breeds, like the Presa Canarios. Some dog breeds have been considered to be “non-modified breeds”, meaning that they retain characteristics more similar to ancestral wolves [36], and they constitute the wolf-like group. These include the German Shepherd based on its skull metrics [37] and in fact, wolves have been said to have a semblance to German Shepherds but smaller [38]. The bite mark patterns registered on dental wax of the adult German Shepherd dog (weight > 25 kg) were analysed recently [39]. These were more than 95% coincident with measurements taken on plaster casts of dog‘s dentition, with all measurements made within five hours to avoid structural changes to the wax. Comparing these results with our outcomes, all the mean values, intervals and maximum values registered for the German Shepherd were greater than those recorded in Iberian wolves for both sexes. However, different overlaps were evident for UCW and LCW intervals of between males of both species, yet only for the UCW in females. Although a direct descendant of Gray wolves, dogs show some morphological differences, such as smaller dentition [20]. Despite rapidly recording the data after registering the bite, we must consider some environmental factors that could possibly influence the results from the wax imprints, like temperature and humidity, as well as those related to the characteristics of the substrate (dental wax in our case), such as the elasticity, hardness, etc.

The dental variables analysed here displayed sexual dimorphism and over 95% of the measurements were larger in males, although some such differences may not have reached statistical significance. This sexual dimorphism was also noted when evaluating dental measures in three different dog breeds [39] and in an earlier study on Iberian wolves [25]. Sexual dimorphism has been mentioned previously when comparing the “anteroposterior diameter of mandibular canines”, corresponding to the Lr/Ll-CLe here [40]. Our data are to some extent consistent with those results, although we found thorough a PLS-DA analysis that these variables are certainly not the only or the most important variables to differentiate the two sexes. Besides, these measurements can be influenced by other factors as discussed below. Sexual dimorphism in Iberian wolves has been studied via dental and cranial dimensions [40] and although the sample size was small, sex differentiation was possible by combining dental and cranial measurements. We found measurements like UP1PM present significance differences between the sexes, with males displaying a larger snout than females from the canine teeth forwards. Other dental measurements also show sexual dimorphism [41], although the utility of these measurements to differentiate sexes has been questioned [42]. In accordance with our nomenclature, they registered very similar mean values for males and females as those presented here for the UrCLe, UlCLe, LlCLe, and LrCLe, even though their measurements were taken at different points of the canines. Other measurements they analysed also showed similar sex differences as those seen here (e.g., the UiW and UCW), such as when comparing the distance separating both the maxillary PM4 (carnassial teeth [40]) with the maxillary distance between the PM4s cusp tip (U4PMc) measured here, producing similar results. Nevertheless, there is some overlap for certain variables between sexes, as seen here, and some ambiguity may exist when morphological characteristics are not due to a given sex or age but rather, reflect individual asymmetry for example (see below).

There is considerable and extensive information about the ecology, conservation and anatomical aspects of Iberian wolves [25]. Anatomically, seven variables on Iberian wolf skulls have been measured, corresponding to UCW, UCc, UbC, LCc, LCW, LCc, and LbC in our study. For the UCW, intervals of 4.5–5 cm were found in males and 3.5–4 cm in females, while the measurements for the UCc were 3–5 mm less than the UCW, and the UBC and LbC were 1.5–2 cm less than the UCW. This study also indicated the measurements for the mandibular canine teeth were 15–20% lower than those from the maxillary arch. Here, we also found lower mandible values than maxillary measurements but we registered a slightly wider variation for these variables (13–20% less in mandible). Greater differences (46–50%) were found between other variables analysed in our study than in this earlier study [25]. The quantification of more dental morphometric variables and their PLS-DA analysis made it possible to graphically display the behaviour of the variables analysed. Despite the mild overlap between sexes and the homogeneous dispersion of variables in each group, the PLS-DA analysis identified two well-defined groups (sexes), as also evident by comparing defined limits to differentiate sexes for each variable. In the light of the results presented here, considering only a few dental dimensions to differentiate sexes may be inaccurate. Thus, it is necessary to process as much data as possible to reduce uncertainty when differentiating sexes. Complementary analysis should determine the most important variables to differentiate sexes and how they influence each other. Wolves and domestic animals are known to exhibit sexual dimorphism [20,43], although to our knowledge, the present study is the first to analyse such extensive morphometric dental data through multivariate methods in order to differentiate the sexes of Iberian wolves.

A degree of asymmetry was evident between values registered on the right and left maxillary and mandibular bones. On the one hand, this could explain why some variables for particular teeth may not be useful to differentiate sexes, generating overlap between the sexes, while on the other hand, these asymmetries may also be reflect sex differentiation. Either way, this asymmetry provides a unique dental morphology for each animal that would permit, if necessary, the identification of any specific animal through its unique dental morphology, as also seen previously [44]. In many cases, a simple tooth can be used for identification if it contains enough unique characteristics [45]. Therefore, morphometric and morphological studies could be combined in a complementary manner for better identification. It has been pointed out that the wear of teeth progresses with age, affecting all teeth [15,46] and especially incisor teeth, the teeth most likely to suffer breaks. This effect was seen in our morphometric study but most specifically, in the mandibular incisor teeth. We noted that all the mandibular incisor teeth in our skulls had a high degree of wear, and if they were absent or fractured, they were not included in our study. The degree of wear of teeth is indicative of the intensity and character of tooth use [15,46], and both these studies indicated that deterioration of a wolf‘s teeth may change their bite pattern and their eating habits. Briefly, I1 and I2 have sharp edges that are used like chisels to take small bites when gnawing at bones, cutting away pieces of food, and in grooming. The canines and I3s serve for deep penetration into the prey’s tissues and for cutting through the tissues, while the premolars strip away food with a cutting action, except for the PM1s (not functional). The small crowns and weakly developed roots of these teeth probably render them useless to hold and carry parts of prey. The molars are designed to crush and pulverize bones to expose the marrow [46], except for the mandibular M3 that is not functional. Therefore, we assume that the roles played by these teeth may be the source of the different degrees of asymmetry found in our study, although they may simply be due to individual characteristics, for example the specimens with outlier values. Despite the asymmetries, two different groups could be defined, namely males and females.

The data suggest that the crowns and the base (cementoenamel junction) of the I3s, canines and PM1s are points of minimal wear. Therefore, variables measured at these points should be more stable over time and more useful to differentiate between sexes. In fact, all the variables measured between both sides of the maxillary and mandible bones, and the variables measured at the cementoenamel junction (UCW, UCc, UbC, LCW, LCc, LbC, UiW, Uic, Ubi, U1PMc, Ur4PMtub, Ul4PMtub, Lr1Mtub, and Ll1Mtub, UrCwd, UrCLe, UlCWd, UlCLe, LrCLe, LlCLe) appeared to be extremely well-suited to differentiate between the sexes. By contrast, the width of the right and left mandibular canine teeth (LrCWd and LlCWd, respectively) was not useful for sexual differentiation, and the L1PMc did not differ significantly between the sexes. Both the maxillary and mandibular PM1s are considered rudimentary teeth, not functional teeth, and they commonly break [15]. In fact, the teeth that most often absent were the maxillary and mandibular PM1s [47]. Here, over 50% of the skulls lacked the L1PM and this reduced number of observations for L1PM may explain why this variable did not differentiate between sexes. Alternatively, we considered that the LrCwd and LlCWd were not suitable to differentiate between sexes due to individual characteristics and not due to the wear of these teeth. In fact, LrCWd and LlCWd showed no significant differences on either side. Contrary to expectations, U4PMc and L1Mc were useful for the sex differentiation, even though the point or tip of their crown is constantly used to crush and pulverize bones. However, we chose accurate cusps and not flat ones, with as little wear as possible. Moreover, according to an earlier study [48], into the wear of teeth the maxillary premolars and M1s in the present study would belong to wolves under six years of age. We must also consider that during occlusion, the cusps of the maxillary premolars and molars do not contact their counterpart on the mandible, producing less wear.

The dispersion of the dental variables analysed here in females was a little higher than in the male group. Similar outcomes were obtained elsewhere when eight mandibular measurements were compared in different kinds of African canids [49]. Here, we used one measurement that was also used in this earlier study, L1Mc, which also proved to be important to differentiate between sexes.

We must remember that some specimens presented values designated as “outliers” in our work, demonstrating high variability for some determined variables, especially in females. The exclusion of these outliers from the morphometric analysis did not cause significant changes in the average values for both sexes, nor did it cause changes in the distribution of the variables within and between the two sexes. However, in other cases these “unusual data” could to alter significantly the distributions. We must also consider the possibility of a degree of reverse dimorphism as an explanation for these outliers, whereby females present higher variability than males. Although Ralls [50] and Reiss [51] listed animal species in which reverse dimorphism could be observed, the only Canidae Family member included was the small-eared zorro (*Atelocynus microtis*). Diverse nutrition, environmental, evaluative, or genetic factors could explain the diversity in sexual dimorphism, as mentioned previously [52], or reverse dimorphism [50,51]. However, we found values considered as outliers for different variables and in different specimens. Moreover, outliers were often asymmetric, with most of them only present on one side (left or right) or in a different jaw (maxillary or mandibular). It would be expected that under normal conditions, genetic, nutritional, evolutive or environmental factors would influence such variables (e.g., teeth) in a more homogeneous way and less asymmetrically. For this reason, we consider that the outlier values obtained here are characteristics of each specimen and they are unlikely to reflect reverse dimorphism. Indeed, the absence of sufficient data regarding the general nature of the sample make it difficult to associate any specific factor to these values.

In Iberian wolves the lateral disposition of the canine teeth’s cusp is important in forensic terms. The depth of the bite will vary in morphological and morphometric terms, with UCW/LCW and UCc/LCc the features most often analysed in bite marks. We see these in superficial bite marks, whereas in a deeper bite, the UbC or LbC may be the most accurate features to measure. Therefore, using intervals could reduce the error due to the depth of the bite. Greater tooth crowding (or smaller diastema sizes) has been seen in Iberian wolves than in the domestic dog, a differentiating characteristic between both canid sub-species [23]. By contrast, some studies have demonstrated that they cannot be confidently separated based on tooth crowding alone [36]. A variation in the length or the absence of diastemas in skull of Iberian wolves was not a consistent characteristic in our study.

A reduction in tooth size has been accepted as the principal signature with which to track domestication [36]. Dog’s teeth are smaller than those of the Iberian wolf [25]. Indeed, the range of sizes among dogs extends much more than that of wolves, distinguishing dogs as the most morphologically diverse terrestrial mammalian species known [53]. For this reason, we must more precisely define the comparisons made in future studies (e.g., wolf-like dogs, wild dogs, domestic, breeds, age, size, etc.).

This study allowed the Iberian wolf’s dentition to be characterized scientifically in morphological and morphometric terms. In a forensic context, the morphological characterization of teeth, including those which had not been morphometrically analysed, will permit their recognition should some of the teeth fall out during the attack or biting process. The intervals defined may be useful to characterize bite marks left on inanimate and/or biological substrates, which have little deformation, such as skeletal or bone remains. The bite marks found may be generated during an attack or in defence, or when scavenging. Their characteristics would also depend on other factors, such as the muscle mass of the victim, the number of animals eating or attacking, the time available for scavenging, etc. [23]. The high variability among dog breeds can lead to a high degree of overlap, such that the use of intervals would be a better option than the average values to compare bite mark patterns in order to determine the guilty party. We have already mentioned some articles in which attempts were made to differentiate and identify diverse families or the genus of carnivorous mammals based on an analysis of the bite marks found on bone remains. The results were mixed, with a wide degree of overlap among them and no studies reporting clear measurements among those patterns. Such analyses become more difficult when bite mark patterns are registered on other biological substrates like skin, which allow significant deformation. To differentiate the same gender in two sub-species is still more difficult, as seen for the Iberian wolf and domestic dog. The differentiation of sexes may be important from an archaeological, population and conservationist point of view, among others. Our study suggest that it should be possible to differentiate the bite marks generated by a male or female Iberian wolf, especially on hard surfaces (as indicated above). In forensic terms, this differentiation can give us important information about the possible ethological nature of an attack (reproductive season?, hormonal factors?, competition for food sources?, etc…). Our data can help to incriminate or exculpate to the Iberian wolf as the possible aggressor against false accusations. Forensic odontology is an important discipline, although caution must be exerted when interpreting the results. To establish a clear separation between sub-species thorough dental morphometry can be risky. Although only some teeth act during biting, a dental analysis must incorporate as many of the variables as possible in order to differentiate both sub-species. As indicated above, ethological, environmental, and population issues must also be considered in forensic investigations. For this and other reasons, the analysis of animal’s bite marks must involve multidisciplinary teams from different forensic areas, such as biologists, veterinarians, dentists, pathologists, and police officers.

The data obtained in this study marks another step in the differentiation between bite mark patterns caused by female and male Iberian wolves. These results would be useful as a base complementary forensic tool for future studies that allow discriminating the Iberian wolf from other sub-species of the Canidae family, such as domestic dogs. New and more standardized and comparison studies must be carried out among Iberian wolves’ dentition vs. wolf-like dogs and large mesocephalic dog breeds, two subspecies that have received little blame without any scientific evidence.

## 5. Conclusions

This study defines the general morphological characteristics of the Iberian wolf dentition in terms of the dental formula and tooth structure. The Iberian wolf presents evident sexual dimorphism in its morphometric dental features. Variables measured on less worn points of teeth are useful in sex differentiation, despite some asymmetry. Therefore, it should be possible to differentiate the bite marks from a male and female Iberian wolf registered on hard/inanimate surfaces with low capacity of deformation (e.g., bones).

Both dental morphometric and morphological analysis are complementary forensic tools, and these features may help identify the Iberian wolf as a suspected aggressor in a more effective and reliable way.

The use of intervals instead of average values seems to be more useful to diferentiate Iberian wolves’ tooth marks between sexes.

Dental characteristics can be used along with other evidence obtained at a site of attack to better interpret the events that occurred.

Given the large number of conflicts caused by the alleged attack of Iberian wolves on domestic cattle, this study’s results may serve as a base complementary forensic tool for future studies. It would ideally allow discriminating the Iberian wolf from other carnivores with similar phenotypic characteristics by means of further comparative studies using bite mark patterns.

## Figures and Tables

**Figure 1 animals-10-00975-f001:**
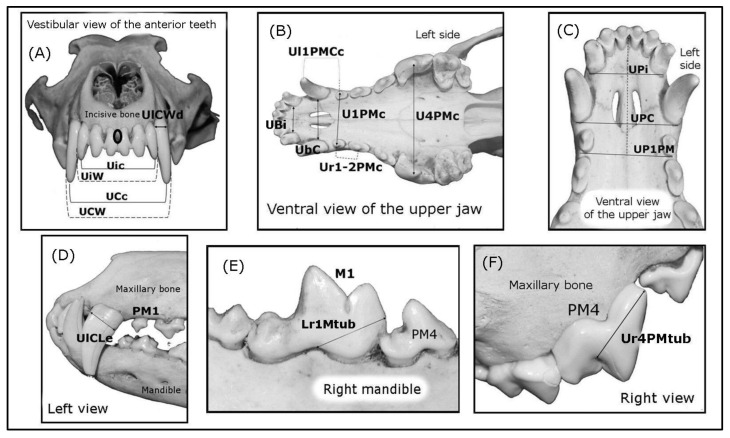
The studied measurements. (**A**) The canine and incisor measurements on the maxillary and incisive bone. The same canine measurements were taken from mandibular canines; (**B**) Measurements were taken on the maxillary dental arch between one tooth and its counterpart on the opposite side, and some were also taken on the mandible; (**C**) Distance from a line drawn behind the three teeth (third maxillary incisors, canines and first premolars) and a point located between both the central incisor teeth on the rostral face of the incisive bone (black circle); (**D**) ULCLe, also registering the same measurement on the right side and for the mandibular canines, (PM1: first premolar); (**E**) Lr1Mtub and the same measurement taken on the left side, (PM4: fourth premolar, M1: first molar); (**F**) Ur4PMtub and the same measurement taken on the left side.

**Figure 2 animals-10-00975-f002:**
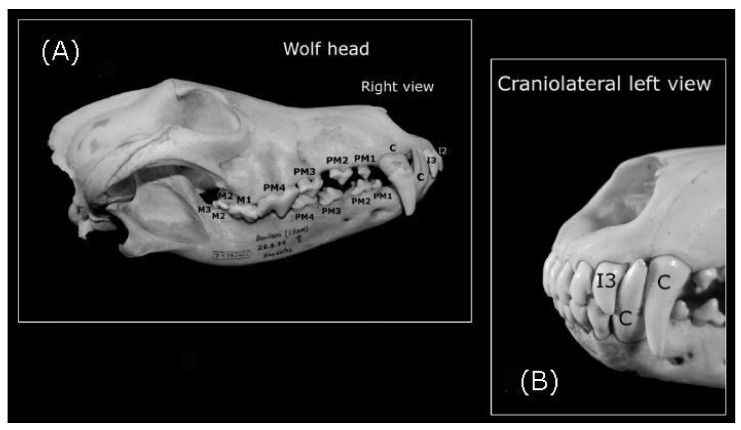
General vision of the Iberian wolf‘s teeth. C: Canine, PM2 and PM3: second and third premolar teeth, respectively; M2 and M3: second and third molar teeth, respectively. (**A**) Wolf skull with the right dental line indicated on the maxillary and mandibular bones; (**B**) Detail of the left-hand canine and I3 on the maxillary and mandibular bones.

**Figure 3 animals-10-00975-f003:**
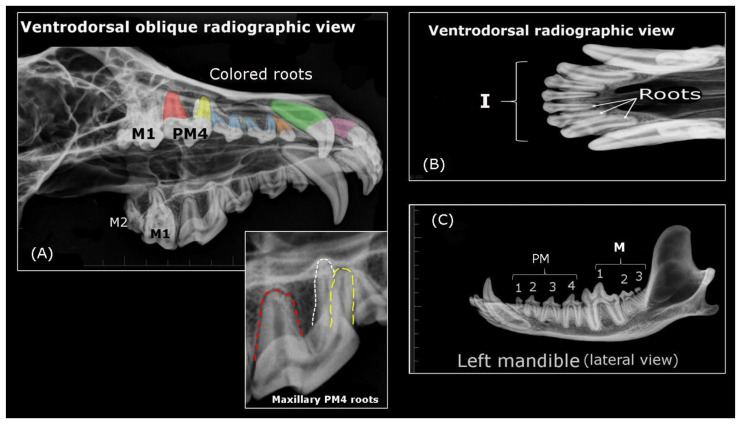
Radiographic views of the wolf‘s teeth and skull. (**A**) (Upper image) Right I3 root (purple), right canine root (green), PM1 root (orange), PM2 and PM3 roots (blue) and PM4 roots (white, red and yellow); (Lower image). Detail of the PM4 roots-mesiobuccal (yellow dotted line), mesiopalatal (white dotted line) and distal root (red dotted line); (**B**) Roots of three right maxillary incisors; (**C**) Molar and premolar disposition teeth in the mandible.

**Figure 4 animals-10-00975-f004:**
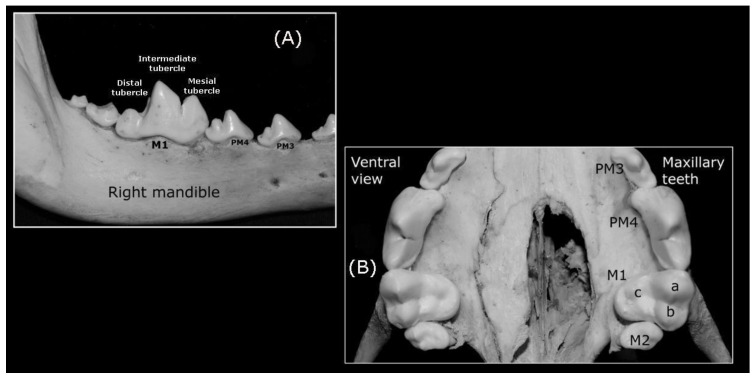
(**A**) Wolf‘s right mandible (lateral view) up to the PM4 tubercles and cusps; (**B**) The occlusive face of left maxillary M1 with its (a) mesiobuccal, (b) distobuccal and (c) distolingual tubercles and cusps.

**Figure 5 animals-10-00975-f005:**
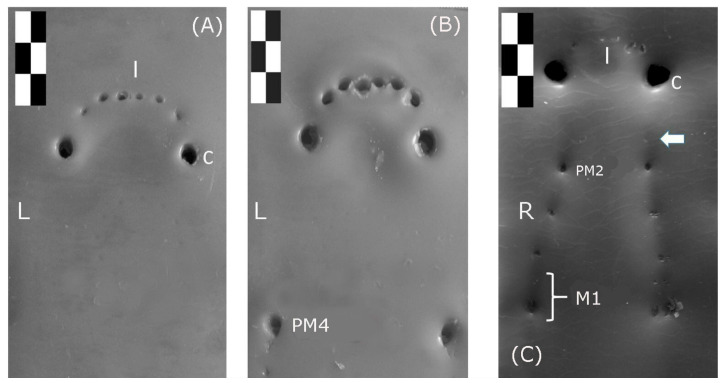
(**A**) Superficial bite mark caused by the maxillary dental arch of the Iberian wolf; (**B**) Deeper bite mark caused by the maxillary dental arch; (**C**) Deep bite mark caused by the mandibular dental arch (mandible). The length of each black/white rectangle at the top left corresponds to one centimetre.

**Figure 6 animals-10-00975-f006:**
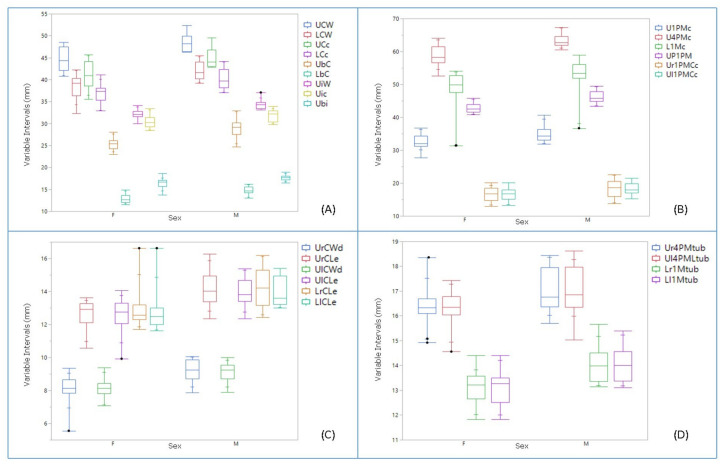
Intervals of the dental measurements considering sex as a source of variation. The minimum, maximum, mean, and quartiles values for each variable according to sex. The black points are the outlier values. (**A**) Measurements taken between right and left canine and between right and left incisors on maxillary and mandibular bones; (**B**) Measurements taken between right and left premolars and between molars; (**C**) Measurements taken on canines, (**D**). Measurements taken on premolars and molars teeth.

**Figure 7 animals-10-00975-f007:**
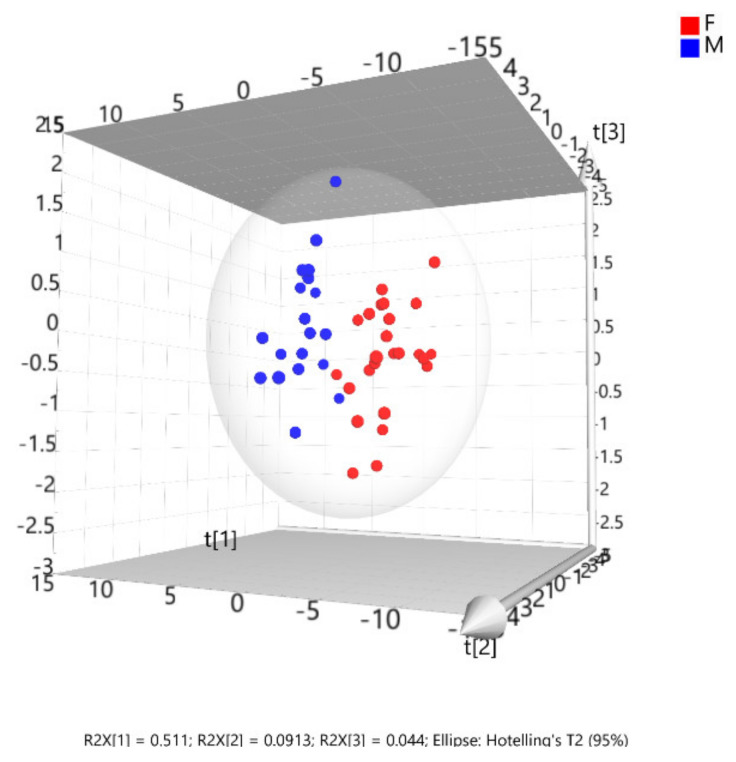
The 3D scatter PLS-DA plot showing the sex differentiation. The 3D scatter PLS-DA plot of the scores showing the distribution and relationship of the variables within and between sexes. The initial latent variables t(1), t(2) and t(3), explained 51.1, 9.13 and 4.4% of the variability, respectively. The whole model explained 64.6 % of the variance.

**Figure 8 animals-10-00975-f008:**
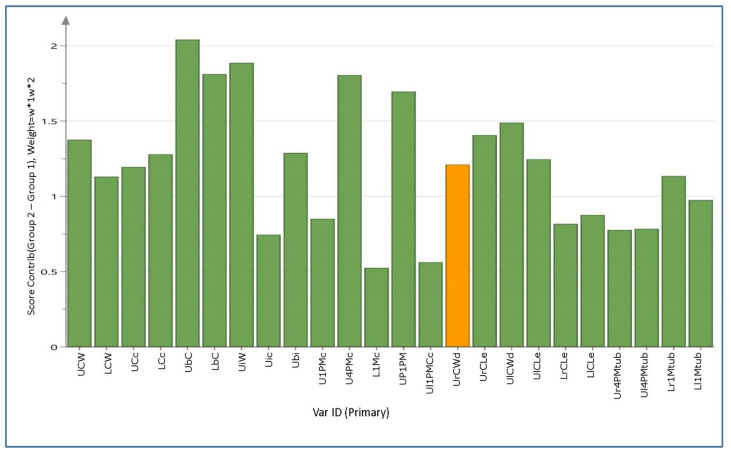
Contribution variables for the sex differentiation. Contribution plot indicating the most significant variables in differentiating males from females with the strongest mean differences (*p* < 0.05). Since the data was a scaled unit vector, the vertical scale units were presented in terms of standard deviations (SDs). The dominant bars show which variables deviate most from the reference point (in this case, the average). The orange colour shows which variables lie outside the 3 SD range, i.e., that which is most influential.

**Table 1 animals-10-00975-t001:** Acronyms of the studied variables.

Variable	Acronym	Variable	Acronym
Maximum maxillary (upper) intercanine width	UCW *	Maximum mandibular (lower) intercanines width	LCW
Maxillary distance between the cusp tip of the canine teeth	UCc *	Mandibular distance between the cusp tip of the canine teeth	LCc
Maxillary distance between the palatal surfaces of the canine teeth	UbC *	Mandibular distance between the lingual surfaces of the canine teeth	LbC
Maximum maxillary inter-third incisor teeth width	UiW *	Maxillary distance between the cusp tip of the third incisor teeth	Uic *
Maxillary distance between the palatal surfaces of the third incisor teeth	Ubi *	Maxillary distance between the cusp tip of the 1st premolar teeth	U1PMc
Mandibular distance between the cusp tip of the 1st premolar teeth	L1PMc	Maxillary distance between the cusp tip of the 4th premolar teeth	U4PMc *
Mandibular distance between the cusp tip of the 1st molar teeth	L1Mc	Distance between a line drawn behind the third maxillary incisors and a point located between both central incisor teeth on the rostral face of incisive bone	UPi *
Distance between a line drawn behind of third maxillary canines and a point located between both central incisor teeth on the rostral face of incisive bone	UPC *	Distance between a line drawn behind maxillary 1st premolars and a point located between both central incisor teeth on the rostral face of the incisive bone	UP1PM *
Right maxillary distance between the cusp tip of the 1st premolar–canine teeth	Ur1PMCc	Left maxillary distance between the cusp tip of the 1st premolar–canine teeth	Ul1PMCc *
Right mandibular distance between the cusp tip of the 1st premolar–canine teeth	Lr1PMCc	Left mandibular distance between the cusp tip of the 1st premolar–canine teeth	Ll1PMCc
Right maxillary distance between the cusp tip of the 1st–2nd premolar teeth	Ur1-2PMc *	Left maxillary distance between the cusp tip of the 1st–2nd premolar teeth	Ul1-2PMc
Right mandibular distance between the cusp tip of the 1st–2nd premolar teeth	Lr1-2PMc	Left mandibular distance between the cusp tip of the 1st–2nd premolar teeth	Ll1-2PMc
Width of the right maxillary canine teeth	UrCWd	Length of the right maxillary canine teeth	UrCLe
Width of the left maxillary canine teeth	UlCWd *	Length of the left maxillary canine teeth	UlCLe *
Width of the right mandibular canine teeth	LrCWd	Length of the right mandibular canine teeth	LrCLe
Width of the left mandibular canine teeth	LlCWd	Length of the left mandibular canine teeth	LlCLe
Length of the right maxillary 4th premolar mesial tubercle	Ur4PMtub *	Length of the left maxillary 4th premolar mesial tubercle	Ul4PMtub
Right mandibular 1st molar tubercle length(includes intermediate and mesial tubercles)	Lr1Mtub *	Length of the left mandibular 1st molar tubercle(including the intermediate and mesial tubercles)	Ll1Mtub

* Measurements as illustrated in Figure 1.

**Table 2 animals-10-00975-t002:** Absolute limits of the variables for the wolf sex differentiation (95% confidence), considering 26 female and 19 male skulls. Va: variable; Lv: Limit value for the sex differentiation; F: females; M: males; mm: millimetres; F/M: sex differentiation was not possible.

Va	Lv (mm)	Mean	Va	Lv (mm)	Mean
UCW	F < 46.25	44.7	Ur1PMCc	F < 13.76	16.48
	M > 48.55	48.4		M > 20.13	18.4
LCW	F < 39.15	38.66	Ul1PMCc	F < 15.27	16.6
	M > 24.23	41.91		M > 20.13	18.39
UCc	F < 42.84	41.27	UrCWd	F < 7.87	8.28
	M > 45.66	45.01		M > 9.36	9.23
LCc	F < 37.06	36.91	UrCLe	F < 12.34	12.62
	M > 41.07	40.09		M > 13.62	14.2
UbC	F < 24.65	25.39	UlCWd	F < 7.88	8.13
	M > 28.04	29.14		M > 9.37	9.11
LbC	F < 13.06	12.91	UlCLe	F < 12.36	12.82
	M > 14.87	14.71		M > 14.05	13.99
UiW	F < 33.10	32.04	LrCLe	F < 12.44	12.7
	M > 34.16	34.18		M > 14.36	14.27
Uic	F < 29.81	30.46	LlCLe	F < 13	12.48
	M > 33.39	31.84		M > 13.69	M: 14.02
Ubi	F < 16.47	16.38	Ur4PMtub	F/M < 17.43	F: 16.41
	M > 18.61	17.68		M > 17.43	17.02
U1PMc	F < 31.83	32.64	Ul4PMtub	F/M < 7.43	F: 16.32
	M > 36.71	34.94		M > 7.43	17
U4PMc	F < 60.55	58.59	Lr1Mtub	F < 3.15	13.06
	M > 64.08	63.57		M > 4.41	14.04
L1Mc	F/M < 54.11	F: 50.54	Ll1Mtub	F < 13.10	13.09
	M > 54.11	54.28		M > 14.4	14.02
UP1PM	F < 43.35	42.92			
	M > 45.84	46.18			

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
