# Peer review of "A Morphological and Morphometric Dental Analysis as a Forensic Tool to Identify the Iberian Wolf (Canis Lupus Signatus)"

_animals, 2020, doi:10.3390/ani10060975_

Round 1

Reviewer 1 Report

This manuscript analyses the utility of morphological and metric dental characteristics as a forensic tool to identify the Iberian wolf. In general, the manuscript is well written and mostly well-structured and executed. However, there are some weaknesses and specific issues which should be addressed.

*The abstract reads: “[…] the aim of this study was to assess the morphological and morphometric characteristics of the Iberian wolf dentition and to contrast these features with the well-known dental morphology of domestic dogs (Canis lupus familiaris)”. Reading the abstract, it seems that a methodology is developed to differentiate the dentition of the Iberian wolf from domestic dogs; however, this study ONLY analyses the dentition of the Iberian wolf. The morphological and metric data of the dogs are not analysed and later no comparison is made. Why? How can be concluded that dentition is useful to define wolves when no comparison was made between Iberian wolves and animals with similar characteristics (e.g., domestic dogs, Gray wolf, etc.)?

*The authors only describe the dentition of the Iberian wolf; they do not perform a true morphological analysis. Only a descriptive study of the specimens is done, but a qualitative analysis of data was not carried out (e.g., through a classification system, a scoring system, etc.). I do not think that the discussion of morphological analysis is appropriate.

* In material and Methods: “Only skulls belonging to anatomically adult wolves without apparent dental damage were included in the study”. How the adulthood of the specimens was determined? Was estimated their age by some specific skeletal methods or this parameter is known because the specimens belong to an identified collection whose biological parameters are known? More information on the biological profile of the specimens should be provided as these parameters could influence the results of the morphological and metric data of the dental variables analysed in the study.

On the other hand, what does “apparent damage” means? A damage is present or not; it should not depend on the judgement of the observer. Were dental and/or bone anomalies and pathologies of the specimens taken into account (e.g., congenital anomalies of the skull, caries, enamel hypoplasia, dental wear, etc.)? The text specifies that “the mandibular incisive teeth were in poor conditions and thus, they were not considered in this study”. But what about the rest of the teeth? More information on this topic should be provided.

*In Figure 1E: The image corresponds to the front view, not the cranial (top) view. If you refers to the skull, then it would be frontal view (or anterior view, rostral view). If you refers to the teeth, you should say vestibular view of the anterior teeth.

*Lines 299-305: Why these variables are considered “outliers” and the existence of a reverse sexual dimorphism for these variables is not contemplated?

It seems that the study is directed and data that are not “of interest” are excluded. Since it is possibly a case of reverse sexual dimorphism, relatively frequent in the literature, these data should not be excluded from the analysis. I highly recommend the articles of Frayer and Wolpoff (1985) and Plavcan (2001) on the meaning of reverse sexual dimorphism in nature.

Frayer DW, Wolpoff MH (1985) Sexual dimorphism. Annual Reviews of Anthropology 14: 429-473.

Plavcan JM (2001) Sexual dimorphism in primate evolution. Yearbook of Physical Anthropology 44: 25-53.

Author Response

The authors are grateful for the invaluable comments that enriched the present manuscript.

As was requested, all changes are clearly highlighted, using the "Track Changes" function in Microsoft Word. Besides, they are shown in grey for a better visualization (e.g., some minor drafting modifications like changes in nomenclature (in text, tables and figures), inserting or deleting individual sentences, phrases or words).

In old line, implies a given line in the previous manuscript sent back by the editor to be revised by us.

Current line, refers to the position of the changes in the final version.

Reviewer 1

This manuscript analyses the utility of morphological and metric dental characteristics as a forensic tool to identify the Iberian wolf. In general, the manuscript is well written and mostly well-structured and executed. However, there are some weaknesses and specific issues which should be addressed.

Point 1.- *The abstract reads: “[…] the aim of this study was to assess the morphological and morphometric characteristics of the Iberian wolf dentition and to contrast these features with the well-known dental morphology of domestic dogs (Canis lupus familiaris)”. Reading the abstract, it seems that a methodology is developed to differentiate the dentition of the Iberian wolf from domestic dogs; however, this study ONLY analyses the dentition of the Iberian wolf. The morphological and metric data of the dogs are not analysed and later no comparison is made. Why? How can be concluded that dentition is useful to define wolves when no comparison was made between Iberian wolves and animals with similar characteristics (e.g., domestic dogs, Gray wolf, etc.)?

Response 1: Our study was not focused on developing a methodology to differentiate the dentition of the Iberian wolf from domestic dogs. The study totally focused on a broad description of the Iberian wolf dentition while providing new, morphometric dental characteristics, as complete as possible. It aims at collaborating in the correct interpretation of the wolf`s bite marks at the crime scenes. Because the wolf predation can sometimes be confused with that caused by other carnivores, we only made some references to other canids in the discussion

Nonetheless, the objective was worded to make it clearer to the reader.

In the old lines 48-51 (current lines 37-40)

“the aim of this study was to assess the morphological and morphometric characteristics of Iberian wolf dentition. This data collection would serve as a base-point for a more accurate identification of the wolves thorough their bite marks”.

Simple Summary was also reworded.

We also propose new comparison studies:

In the old lines 682 – 683 (current lines 600-601)

“New and more standardized and comparison studies must be carried out among Iberian wolves dentition vs. wolf-like dogs and large mesocephalic dog breeds”

Besides, the Conclusions were reviewed

In the old lines 696 – 697 (eliminated)

“Due to the high variability among the dog breeds, it is difficult and risky to establish a separation between both sub-species using the dental morphometry”.

Finally,

In the old lines 698-699 (current lines 614-615) the text was rephrased as follows:

 “The use of intervals instead of the average values seemed to be more useful to differentiate the Iberian wolves’ tooth marks between sexes”

Point 2- *The authors only describe the dentition of the Iberian wolf; they do not perform a true morphological analysis. Only a descriptive study of the specimens is done, but a qualitative analysis of data was not carried out (e.g., through a classification system, a scoring system, etc.). I do not think that the discussion of morphological analysis is appropriate.

Response 2:  Thanks for the comment. We reworded the non-clear title to be clearer:

In the old line: 151 (current line 118)

2.2. Morphological description

Besides, we have excluded the discussion about a comparative morphological description for dog and wolves’ teeth. However, we stress the importance of the morphological description of the teeth, thus the following text was added:

In the old lines 649-651 (current lines 568-570)

“In a forensic context, the morphological characterization of teeth, including those which had not been morphometrically analysed, will permit their recognition should some of the teeth fall out during the attack or biting process”

Point 3. * In material and Methods: “Only skulls belonging to anatomically adult wolves without apparent dental damage were included in the study”. How the adulthood of the specimens was determined? Was estimated their age by some specific skeletal methods or this parameter is known because the specimens belong to an identified collection whose biological parameters are known? More information on the biological profile of the specimens should be provided as these parameters could influence the results of the morphological and metric data of the dental variables analysed in the study.

On the other hand, what does “apparent damage” means? A damage is present or not; it should not depend on the judgement of the observer. Were dental and/or bone anomalies and pathologies of the specimens taken into account (e.g., congenital anomalies of the skull, caries, enamel hypoplasia, dental wear, etc.)? The text specifies that “the mandibular incisive teeth were in poor conditions and thus, they were not considered in this study”. But what about the rest of the teeth? More information on this topic should be provided.

Response 3: Considering the valuable contributions, we have included changes in Material and Methods and we have provided more information about the biological profile of the specimens used in the present study:

In the old lines 95-101 (current lines 80-85)

“Only wolves skulls identified anatomically with mature dentition (permanent dentition), and with all the teeth at eruption stage 3, were included in the study. This means that all incisors, premolars and molar teeth had fully erupted into the occlusal plane. Likewise, the visible cementoenamel junction was above the alveolus, in accordance with the code used by Geiger et al., [16]. No congenital skull anomalies or external dental damage were evident in the skulls sampled”.

Point 4. *In Figure 1E: The image corresponds to the front view, not the cranial (top) view. If you refers to the skull, then it would be frontal view (or anterior view, rostral view). If you refers to the teeth, you should say vestibular view of the anterior teeth.

Response 4:  We have changed the nomenclature in the figure 1A to the “vestibular view of the anterior teeth”. We refer to the teeth.

Point 5. *Lines 299-305: Why these variables are considered “outliers” and the existence of a reverse sexual dimorphism for these variables is not contemplated?

It seems that the study is directed and data that are not “of interest” are excluded. Since it is possibly a case of reverse sexual dimorphism, relatively frequent in the literature, these data should not be excluded from the analysis. I highly recommend the articles of Frayer and Wolpoff (1985) and Plavcan (2001) on the meaning of reverse sexual dimorphism in nature.

Frayer DW, Wolpoff MH (1985) Sexual dimorphism. Annual Reviews of Anthropology 14: 429-473.

Plavcan JM (2001) Sexual dimorphism in primate evolution. Yearbook of Physical Anthropology 44: 25-53.

Response 5: Some variables and one female skull were considered as outliers because they were outside the 95 % confidence level. Moreover, as we indicated, the exclusion of these variables considered outliers from the statistical analysis affected some intervals. However, there were no significant changes in the average values of males and females, i.e., the relationship between sexes remained stable. To clarify

In the old line 376 (current line 318) we added: 

  “outside the 95 % confidence level”

To graphically support this affirmation we also added complementary information as Supplementary material (Figure S1 and Table S1):

In the old line 380 (current line 322) we added:

“(Figure S1 and Table S1)”. (Supplementary Materials)

On other hand, the exclusion of that one female skull did not cause any variation in the distribution of the variables within and between the two sexes.

To graphically support this affirmation we added new information as Supplementary material (Figure S2)

In the old line 404 (current line 346) we added:

“(Figure S2)” (Supplementary Materials)

Dr. Francisco Ocaña in: Statistical techniques in Nutrition and Health (outliers and missing data) (Spanish book) indicate “..If the observations influence the results, we should report it and try to explain why these observations.”

In our case, as pointed out earlier, their exclusion from the morphometric analysis did not cause significance differences

Reference can be visited in: https://www.ugr.es/~fmocan/MATERIALES%20DOCTORADO/Tratamiento%20de%20outliers%20y%20missing.pdf

Nonetheless, to give an answer about the reverse dimorphism as well as a possible explanation to the outlier values, we reviewed the references given by the reviewer and others. We strongly believe that the asymmetric distribution and presentation shown in different samples were not caused by the factors mentioned in the literature. We believe that these unusual characteristics, associated to the outliers, are specific to each individual caused by factors we ignore. We express this in the following text added in the discussion section.

In the old lines 608-626 (current lines 532-550)

“We must remember that some individuals presented values designated as “outliers” in our work, demonstrating high variability for some determined variables, especially in females. The exclusion of these outliers from the morphometric analysis did not cause significant changes in the average values for both sexes, nor did it cause changes in the distribution of the variables within and between the two sexes. However, in other cases these “unusual data” could to alter significantly the distributions. We must also consider the possibility of a degree of reverse dimorphism as an explanation for these outliers, whereby females present higher variability than males. Although Ralls [50] and Reiss [51] listed animal species in which reverse dimorphism could be observed, the only Canidae Family member included was the Small-eared zorro (Atelocynus microtis). Diverse nutrition, environmental, evaluative or genetic factors could explain the diversity in sexual dimorphism, as mentioned previously [52], or reverse dimorphism [50, 51]. However, we found values considered as outliers for different variables and in different individuals. Moreover, outliers were often asymmetric, with most of them only present on one side (left or right) or in a different jaw (upper or lower). It would be expected that under normal conditions, genetic, nutritional, evolutive or environmental factors would influence such variables (e.g. teeth) in a more homogeneous way and less asymmetrically. For this reason, we consider that the outlier values obtained here are characteristics of each individual and they are unlikely to reflect re dimorphism. Indeed, the absence of sufficient data regarding the general nature of the samples make it difficult to associate any specific factor to these values.

 Language

BiomedRed Company (Dr. Marck Sefton) did the text translation. Website: http://www.biomedred.com/#!/pages/home

Sincerely,

Víctor Toledo González

Reviewer 2 Report

  1. The title of this manuscript suggests and throughout this manuscript the authors refer to a comparison with domestic dogs, the objective being to be able to differentiate bite marks from wolfs from those of domestic dogs. For this purpose morphometric measurements of the dentition of wolves are provided. A major flaw is that the authors have not studied (or do not report) the same morphometric measurements in dogs, so any and all references to domestic dogs are therefore unsubstantiated and inappropriate. Common sense dictates that there is a big difference between the bite marks of a small-medium dolichocephalic dog, such as a whippet, and a brachycephalic dog, such as an English bulldog, but the unanswered and important question remains how to differentiate a large mesocephalic breed, such as a German shepherd, from a wolf. This is not addressed in this manuscript. 2. The authors offer a very detailed comprehensive description of the dentition of the wolf, but this is nothing new. This has all been described decades ago. Moreover, the terminology used is not in accordance with the current Nomina Anatomica Veterinaria and comparative odontology texts. For example, there are no “incisives” and teeth have no “lobes”, and I would assume that “amelocemental joint” probably refers to the “cementoenamel junction”. 3. The results suggest that it should be possible to differentiate the bite marks from a male vs. a female Iberian wolf. Unfortunately, nowhere in the manuscript does it clearly and succinctly state so. 4. Given the critical nature of my review, I have not provided line-by-line editorial comments.

Author Response

The authors are grateful for the invaluable comments that enriched the present manuscript.

As was requested, all changes are clearly highlighted, using the "Track Changes" function in Microsoft Word. Besides, they are shown in grey for a better visualization (e.g., some minor drafting modifications like changes in nomenclature (in text, tables and figures), inserting or deleting individual sentences, phrases or words).

In old line, implies a given line in the previous manuscript sent back by the editor to be revised by us.

Current line, refers to the position of the changes in the final version.

Point 1. The title of this manuscript suggests and throughout this manuscript the authors refer to a comparison with domestic dogs, the objective being to be able to differentiate bite marks from wolfs from those of domestic dogs. For this purpose morphometric measurements of the dentition of wolves are provided. A major flaw is that the authors have not studied (or do not report) the same morphometric measurements in dogs, so any and all references to domestic dogs are therefore unsubstantiated and inappropriate. Common sense dictates that there is a big difference between the bite marks of a small-medium dolichocephalic dog, such as a whippet, and a brachycephalic dog, such as an English bulldog, but the unanswered and important question remains how to differentiate a large mesocephalic breed, such as a German shepherd, from a wolf. This is not addressed in this manuscript. 

Response 1: The authors assent to the reviewer’s suggestion. Thus, we restate the answer sent to the previous reviewer about this point.

Our study was not focused on developing a methodology to differentiate the dentition of the Iberian wolf from domestic dogs. The study totally focused on a broad description of the Iberian wolf dentition while providing new, morphometric dental characteristics, as complete as possible. It aims at collaborating in the correct interpretation of the wolf`s bite marks at the crime scenes. Because the wolf predation can sometimes be confused with that caused by other carnivores, we only made some references to other canids in the discussion

Nonetheless, the objective was worded to make it clearer to the reader.

In the old lines 48-51 (current lines 37-40)

“the aim of this study was to assess the morphological and morphometric characteristics of Iberian wolf dentition. This data collection would serve as a base-point for a more accurate identification of the wolves thorough their bite marks”

Simple Summary was also reworded

We also propose new comparison studies:

In the old lines 682 – 683 (current lines 600-601)

“New and more standardized and comparison studies must be carried out among Iberian wolves dentition vs. wolf-like dogs and large mesocephalic dog breeds”

Point 2. The authors offer a very detailed comprehensive description of the dentition of the wolf, but this is nothing new. This has all been described decades ago. Moreover, the terminology used is not in accordance with the current Nomina Anatomica Veterinaria and comparative odontology texts. For example, there are no “incisives” and teeth have no “lobes”, and I would assume that “amelocemental joint” probably refers to the “cementoenamel junction”. 

Response 2: It is possible to recover scanty and diverse information about certain characteristics mentioned in scientific articles or books with editorial committee regarding the wolves` dentition morphology. However, most of them refer only to dental formula, eruption process, functional aspects and, pathology. In addition, most of them denote American wolves and just a few mention Iberian wolves. For those reason, we already state the following:

In the old line 478-479 (current line 404-405)

”…the average size of the Iberian wolf lies somewhere between that of the American and European wolves (25)” so new and complete information is required.

On the other hand, diverse descriptions are made using wolves`s bite mark patterns or patterns of bone modifications on heterogeneous samples, specially, in a paleoecological and zooarchaeological context as we mentioned in:

In the old and current line 25

But there is still limited information on Iberian wolf`s dental anatomy that can be used in forensic cases as we mentioned in:

In the old and current line 26

Focusing on this context for futures studies, we need to provide new morphometric characteristics, as complete as possible, to collaborate in the correct interpretation of the wolf`s bite marks at the crime scenes as mentioned in:

In the old and current line 27

To support this explanation, some references are given as follows:

Haynes, G. 1980, 1983; Binford, L.R 1981; D’Andrea and R. Gotthardt, 1984; Andrews, 1995; Haglund, W.D. and M.H. Sorg, 1997;  Yravedra et al. 2011, 2013, 2014, 2017, 2019; Andrés et al., 2012; Fosse et al. 2012; Domínguez-Rodrigo et al., 2012; Burke C. 2013; Parkinson, J. et al., 2014; Nohemi Sala et al. 2014; Lescureux, N. and J. Linnell 2014;  Julia Aramendi et al. 2017; Iglesias, A., A. España, and J. España, 2017;  AE Pires et al., 2020.

This study implied reviewing more than 500 articles regarding wolves’ bite marks patterns from another journals of forensic science. The authors do not know about another scientific publication or book with editorial committee which describes the Iberian wolf`s dentition as complete as this work.

Finally, the terminology or some anatomical terms were corrected based in Oral anatomy and physiology. In Wigg`s veterinary dentistry. (2019)

In the old line 154 (current line 121)

These changes can be clearly founded, using the "Track Changes" function in Microsoft Word. Besides, they are shown in grey for a better visualization

Point 3. The results suggest that it should be possible to differentiate the bite marks from a male vs. a female Iberian wolf. Unfortunately, nowhere in the manuscript does it clearly and succinctly state so. 

Response 3: Besides/In addition, in the Abstract we informed about the  possibility to differentiate both sexes using the morphometric analysis of the Iberian wolves` teeth  in:

In the old line 57 (current line 43)

Sex differentiation was evident, principally in terms of…”

We considered to reinforce this point rewording the text as follows:

In the old line 667-669 (current line 585-586)

“Our study suggests that it should be possible to differentiate the bite marks generated by a male or female Iberian wolf”

And

In the old lines 698-699 (current lines 614-615)

“The use of intervals instead of the average values seems to be more useful to differentiate the Iberian wolves` tooth marks between sexes”.

Point 4. Given the critical nature of my review, I have not provided line-by-line editorial comments

Language

BiomedRed Company (Dr. Marck Sefton) did the text translation. Website: http://www.biomedred.com/#!/pages/home

Sincerely,

Víctor Toledo González

Reviewer 3 Report

The subject of the article is very relevant and useful. New forensic methodologies are needed to interpret wolf attacks upon livestock.

The fact that this study is not able to use morphometry to distinguish dogs and wolves is a major setback.

Probably currently its not accessible but for future works the authors have a collection of dogs at the Universidad Autónoma de Madrid at the care of Professor Arturo Morales-Muniz. If I remember well breed and sex are registered. Maybe it could be useful to compare.

Minor changes:

Line 24: it should be Paleoecology an Zooarhcaeology

Line 84: Refer also to figure 1

Author Response

The authors are grateful for the invaluable comments that enriched the present manuscript.

As was requested, all changes are clearly highlighted, using the "Track Changes" function in Microsoft Word. Besides, they are shown in grey for a better visualization (e.g., some minor drafting modifications like changes in nomenclature (in text, tables and figures), inserting or deleting individual sentences, phrases or words).

In old line, implies a given line in the previous manuscript sent back by the editor to be revised by us.

Current line, refers to the position of the changes in the final version.

Reviewer 3

Point 1. - The subject of the article is very relevant and useful. New forensic methodologies are needed to interpret wolf attacks upon livestock.

The fact that this study is not able to use morphometry to distinguish dogs and wolves is a major setback.

Response 1:  The authors assent to the reviewer’s suggestion. Thus, we restate the answer sent to the previous reviewer about this point.

Our study was not focused on developing a methodology to differentiate the dentition of the Iberian wolf from domestic dogs. The study totally focused on a broad description of the Iberian wolf dentition while providing new, morphometric dental characteristics, as complete as possible. It aims at collaborating in the correct interpretation of the wolf`s bite marks at the crime scenes. Because the wolf predation can sometimes be confused with that caused by other carnivores, we only made some references to other canids in the discussion

Nonetheless, the objective was worded to make it clearer to the reader.

In the old lines 48-51 (current lines 37-40)

“the aim of this study was to assess the morphological and morphometric characteristics of Iberian wolf dentition. This data collection would serve as a base-point for a more accurate identification of the wolves thorough their bite marks”

Simple Summary was also reworded

We also propose new comparison studies:

In the old lines 682 – 683 (current lines 600-601)

“New and more standardized and comparison studies must be carried out among Iberian wolves dentition vs. wolf-like dogs and large mesocephalic dog breeds”

Point 2.- Probably currently its not accessible but for future works the authors have a collection of dogs at the Universidad Autónoma de Madrid at the care of Professor Arturo Morales-Muniz. If I remember well breed and sex are registered. Maybe it could be useful to compare.

Minor changes:

Line 24: it should be Paleoecology and Zooarhcaeology

Line 84: Refer also to figure 1

Response 2: We started to work in this point. Unfortunately, we are stopped for global situation. For future studies, I will contact to Professor Arturo Morales-Muniz. Thank you very much.

Old Line 25 (current line 25) and old line 490 (current 415) were changed to

“Paleoecological and Zooarchaeological”

Language

BiomedRed Company (Dr. Marck Sefton) did the text translation. Website: http://www.biomedred.com/#!/pages/home

Sincerely,

Víctor Toledo González

Round 2

Reviewer 1 Report

The authors have addressed the comments and suggestions offered by the reviewers and the manuscript has significantly improved. In the following please find some relatively minor and specific suggested changes to assist with the ease in reading.

Specific notes/changes:

Line 25: If use British English, change “Paleoecological” to “Palaeoecological”. Please, do not use mixed British/American English. Check within the text for mixed British/American English vocabulary.

Line 99: change “with” to “with” (the letter ‘t’ is in italics).

Line 218: change “incisives” to “incisors”.

Lines 399, 580, 608, 619, 624: change “individual(s)” to “specimen(s)”.

Line 624: change “re dimorphism” to “reverse dimorphism”.

Line 625: change “samples” to “sample” or “specimens”.

Line 629: check this sentence (the term “teeth” must be removed????)

Lines 402–404, 407, 486: change “sample(s)” to “specimen(s)”.

Line 448: change “[…] 14 indeterminate:” to “[…] 14 indeterminate sex).”. Add the term ‘sex’ and change “:” to “).”

Lines 461, 468: change “Canis familiaris” to “Canis familiaris” (in italics).

Lines 501–507: change “German shepherd” to “German Shepherd”.

Line 501: “skull osteometry” means skull metrics??? In this case, change “skull osteometry” to “skull metrics” or “skull size”.

Line 516: change “This dimorphism” to “This sexual dimorphism”.

Within the text and figures: to keep consistent with dental nomenclature, use the same nomenclature. Please, select PM4, fourth premolar or 4th premolar or 4th premolar; M2 or second molar or 2nd molar or 2nd molar; and so on. Do not mix different dental nomenclatures within the text and figures.

Author Response

Cover letter (revisions and responses to the reviewer' comments)

The authors are grateful for the invaluable comments that enriched the present manuscript.

As was requested, all changes are clearly highlighted, using the "Track Changes" function in Microsoft Word. Besides, they are shown in grey for a better visualization (e.g., some minor drafting modifications like changes in nomenclature (in text, tables and figures), inserting or deleting individual sentences, phrases or words).

Point 1: Line 25:  If use British English, change “Paleoecological” to “Palaeoecological”. Please, do not use mixed British/American English. Check within the text for mixed British/American English vocabulary.

Response 1: (Current line 25, in clear version):  Changed Paleoecological” to “Palaeoecological”

Point 2: Line 99: change “with” to “with” (the letter ‘t’ is in italics).

Response 2: changed “with” to “with (Current line 84, in clear version)

Point 3: Line 218: change “incisives” to “incisors”. (Line 183 in clear version)

Response 3: changed “incisives” to “incisors”. (Current line 183, in clear version)

Point 4: Lines 399, 580, 608, 619, 624: change “individual(s)” to “specimen(s)”.

Response 4: Changed “individual(s)” to “specimen(s).  (Current lines 342, 508, 536, 547 and 552, in clear version)

Point 5: Line 624: change “re dimorphism” to “reverse dimorphism”.

Response 5: Changed “re dimorphism” to “reverse dimorphism”. (Current lines 552, in clear version)

Point 6: Line 625: change “samples” to “sample” or “specimens”.

Response 6: Changed “samples” to “sample” (Current line 553, in clear version)

Point 7: Line 629: check this sentence (the term “teeth” must be removed????)

Response 7:  Totally agree (Current line 556, in clear version)

Point 8: Lines 402–404, 407, 486: change “sample(s)” to “specimen(s)”.

Response 8: It was a little bit difficult to find them in new clear version. I believe I did all the required changes: (Current lines 345-347 and 394, in clear version).

Point 9: Line 448: change “[…] 14 indeterminate:” to “[…] 14 indeterminate sex).”. Add the term ‘sex’ and change “:” to “).”

Response 9: changed to: indeterminate sex (Current line 375, in clear version)

Point 10: Lines 461, 468: change “Canis familiaris” to “Canis familiaris” (in italics).

Response 10: changed to italic: (Current lines 388, 395, 741). Other scientific names in references were changed to italic. All changes are clearly highlighted, using the "Track Changes" function in Microsoft Word. Besides, they are shown in grey for a better visualization

Point 11: Lines 501–507: change “German shepherd” to “German Shepherd”.

Response 11: Changed in Current lines 428, 430 and 434, in clear version

Point 12:  Line 501: “skull osteometry” means skull metrics??? In this case, change “skull osteometry” to “skull metrics” or “skull size”.

Response 12: Changed “skull osteometry” to “skull metrics”, in order to avoid redundancy (Current line 428, in clear version)

Point 13: Line 516: change “This dimorphism” to “This sexual dimorphism”.

Response 13: We changed to “This sexual dimorphism” (Current line 444, in clear version)

Point 14: Within the text and figures: to keep consistent with dental nomenclature, use the same nomenclature. Please, select PM4, fourth premolar or 4th premolar or 4th premolar; M2 or second molar or 2nd molar or 2nd molar; and so on. Do not mix different dental nomenclatures within the text and figures.

Response 14: Dental nomenclature was standardized in the text and figures.

To specific dental names: PM1, Pm2, Pm3 and so on.

General dental names: Incisors, canines, premolars and molars.

In addition, the terms as “upper” and “lower”-(teeth/ tooth/ dental arch, etc.) were replaced to “maxillary” and “mandibular”, respectively.

All changes are highlighted, using the "Track Changes" function in Microsoft Word. Besides, they are shown in grey for a better visualization

Sincerely,

Victor Toledo G.

Reviewer 2 Report

Thank you for making many of the suggested changes. The manuscript is much approved. My main concern remains that the most important question, namely the ability to differentiate the bite mark of a wolf and a large mesocephalic dog, has not been addressed.

Author Response

The authors are grateful for all the comments; they are helping to improve the final manuscript.

As was requested, all changes are clearly highlighted, using the "Track Changes" function in Microsoft Word. Besides, they are shown in grey for a better visualization (e.g., some minor drafting modifications like changes in nomenclature (in text, tables and figures), inserting or deleting individual sentences, phrases or words).

Point 1. My main concern remains that the most important question, namely the ability to differentiate the bite mark of a wolf and a large mesocephalic dog, has not been addressed

Response: In text, we have indicated in current lines 605- 608 (clear revision)

 “New and more standardized and comparison studies must be carried out among Iberian wolves dentition vs. wolf-like dogs and large mesocephalic dog breeds, two subspecies that have received some blame without any scientific evidence”.

We are aware of the inexistence of detailed and comparative studies in forensic context among phenotypically similar species. We also aware of the inexistence of detailed and comparative studies in forensic context. For this reason, the corresponding discussion section was reworded as follows:

Current lines 415-421

 “Most of these past studies on bite marks, focused principally on archaeological and zooarchaeological samples, and contexts, with ambiguous outcomes. This represented a huge effort undertaken by those authors whom worked with non-standardized samples, exposed to unknown external conditions, which may have caused the results variability they obtained. Indeed, they focused on bite mark characteristics caused by only a few teeth. To complement and extend these studies, we collected here as many dental variables as possible in order to make it possible to identify the bite mark patterns caused by Iberian wolves in a forensic context”. 

Besides, in order to clarify the planed and indicated objectives of this study, a few lines of the conclusions were reworded as follows:

Current lines 610 -611

 “This study defines the general morphological characteristics of the Iberian wolf dentition, in terms of the dental formula and tooth structure”.

Finally,

Current lines 623-626 were replaced by:

“Given the large number of conflicts caused by the alleged attack of Iberian wolves on domestic cattle, this study’s results may serve as a base complementary forensic tool for future studies. It would ideally allow discriminating the Iberian wolf from other carnivores with similar phenotypic characteristics by means of further comparative studies using bite mark patterns”.

We hope we have answered all the questions and issues the reviewers stated.

Sincerely,

Victor Toledo G.
